# Genetic Diversity and Population Structure of the Asian Tiger Mosquito (*Aedes albopictus*) in Vietnam: Evidence for Genetic Differentiation by Climate Region

**DOI:** 10.3390/genes12101579

**Published:** 2021-10-06

**Authors:** Cuong-Van Duong, Ji-Hyoun Kang, Vinh-Van Nguyen, Yeon-Jae Bae

**Affiliations:** 1Department of Environmental Science and Ecological Engineering, College of Life Sciences, Korea University, Seoul 02841, Korea; dvcuong92@korea.ac.kr; 2Korean Entomological Institute, College of Life Sciences, Korea University, Seoul 02841, Korea; jihyounkang@korea.ac.kr; 3Department of Applied Zoology, Faculty of Biology, VNU University of Science, Vietnam National University, Hanoi 100000, Vietnam; vinhvn@hus.edu.vn

**Keywords:** *Aedes albopictus*, genetic diversity, population structure, mitochondrial DNA, Vietnam

## Abstract

*Aedes albopictus* is a native mosquito to Southeast Asia with a high potential for disease transmission. Understanding how *Ae. albopictus* populations that develop in the species’ native range is useful for planning future control strategies and for identifying the sources of invasive ranges. The present study aims to investigate the genetic diversity and population structure of *Ae. albopictus* across various climatic regions of Vietnam. We analyzed mitochondrial cytochrome oxidase I (*COI*) gene sequences from specimens collected from 16 localities, and we used distance-based redundancy analysis to evaluate the amount of variation in the genetic distance that could be explained by both geographic distance and climatic factors. High levels of genetic polymorphism were detected, and the haplotypes were similar to those sequences from both temperate and tropical regions worldwide. Of note, these haplotype groups were geographically distributed, resulting in a distinct population structure in which northeastern populations and the remaining populations were genetically differentiated. Notably, genetic variation among the *Ae. albopictus* populations was driven primarily by climatic factors (64.55%) and to a lesser extent was also influenced by geographic distance (33.73%). These findings fill important gaps in the current understanding of the population genetics of *Ae. albopictus* in Vietnam, especially with respect to providing data to track the origin of the invaded regions worldwide.

## 1. Introduction

Within its native range of Southeast Asia, the Asian tiger mosquito *Aedes albopictus* (Skuse, 1894) is a tree hole-dwelling zoophilic species [1]. However, during the last few decades, the species has rapidly expanded into a variety of ecological niches in both tropical and temperate regions of the Americas, Africa, and Europe [2], facilitated mainly by international trade and travel [3,4]. Furthermore, the species’ ability to reproduce in domestic containers and to undergo diapause during unfavorable climatic conditions are additional characteristics that have allowed *Ae. albopictus* to survive unfavorable conditions or novel environments [5]. Interestingly, the capacity for photoperiodic diapause is only found in *Ae. albopictus* populations in temperate regions, while tropical populations are active all year round [6,7]. Several population genetic studies have suggested the existence of genetic differentiation between distant temperate and tropical populations [8,9,10,11], it remains unclear whether the two groups are genetically differentiated in the species’ native range.

As one of the world’s most medically important vectors, the species is capable of transmitting many arboviruses, including dengue, chikungunya, and Zika [1,12,13,14,15]. Owing to the absence of effective arbovirus vaccines or treatments, the prevention of most mosquito-borne diseases relies predominantly on the reduction of vector populations [16], and effective control strategies can be guided by information about vector species’ population genetics, including sink and source populations, current and future population dynamics, and population scale [17]. However, even though many studies have investigated the population genetics of *Ae. albopictus*, few have included samples from the species’ native range (i.e., southeast Asia), and for this reason, little is known about the origin of invasive populations. 

Additionally, recent studies have demonstrated that geographic distance is not the only factor that affects gene flow. Indeed, the population structure may also be affected by environmental factors, either alone or in conjunction with geographic distance [18]. In particular, major climate factors such as temperature and precipitation are known to affect *Ae. albopictus* dispersal [19]. However, few, if any, studies have investigated the associations between the species’ genetic variation and the effects of geographic distance and climatic factors. 

Vietnam is known as one of the native ranges of *Ae. albopictus* and is suspected as the source of documented invasions into several tropical nations, including Cameroon, Brazil, and Madagascar [20,21,22]. Thus, investigating the population genetics of the species in Vietnam is essential to improving the current understanding of the geographic origins of past invasions and, thus, to facilitate the development of control strategies. Furthermore, Vietnam has a tropical monsoon climate, but northern Vietnam is influenced by the northeast monsoon and experiences a temperate or subtropical climate, whereas southern Vietnam typically experiences a tropical climate [23]. Moreover, due to the effects of natural mountainous boundaries and monsoon activities, Vietnam’s climate can be separated into seven climatic regions [24]. *Aedes albopictus* occurs throughout Vietnam [25] and is adapted to a variety of environments, which could potentially cause gradual changes in the morphology, behavior, and genetics of respective populations [26]. Given those conditions, Vietnam is an ideal land to examine the effects of climatic heterogeneity to genetic variation and population structure of *Ae. albopictus*. In addition, Vietnam has always ranked among the countries most heavily affected by mosquito-borne diseases [27,28,29,30]. Dengue fever, which is the most serious disease transmitted by *Aedes aegypti* (Linnaeus, 1762) and *Ae. albopictus*, represents a major threat to both the public health and economy of Vietnam [31,32]. Given the value of population genetic information for establishing vector control programs, the genetic analysis of *Ae. albopictus* populations is essential to managing the further spread of dengue fever in Vietnam. However, the genetic availability of *Ae. albopictus* populations in its native range in Vietnam has not been conducted yet. 

The present study aims to evaluate the variability of mitochondrial cytochrome oxidase I (*COI*) gene sequences from *Ae. albopictus* populations in Vietnam, to investigate the population structure of the species among temperate and tropical regions, and to evaluate the role of geographic distance and climatic factors in determining genetic differentiation. The study’s findings are expected to improve the current understanding of the geographic origins of recent *Ae. albopictus* invasions and to provide evidence for ongoing gene flow restriction between the species’ temperate and tropical populations. The implications of these findings for mosquito control in Vietnam are also discussed.

## 2. Materials and Methods

### 2.1. Insect Materials

From June to October 2019, mosquito specimens were collected from 16 sites that were distributed across a north–south transect in Vietnam (Figure 1A, Appendix A) and that were associated with seven climatic regions: northwest (NW), northeast (NE), north delta (ND), north central coast (NC), south central coast (SC), central highlands (CH), and the south (S) [24]. 

At each site, larvae and/or adult mosquitoes (irrespective of sex) were collected from a variety of habitats, including domestic containers, discarded tires and containers, bonsai tanks, flowerpots, and tree holes. Adults were collected using aspirators and larvae were collected using a small net and forceps. Larvae were either allowed to mature under laboratory conditions or preserved in absolute ethanol in the field. All specimens were initially identified morphologically using taxonomic keys [33] and were then stored in absolute alcohol at −20 °C for subsequent genetic analysis. To avoid the analysis of siblings, only one or two specimens of neither adult nor larva from each source at individual sites were used for genotyping. Special permissions were not required for the collection of mosquitoes at any of the sites, and in the case of private residential areas, permission was obtained from owners before entry.

### 2.2. DNA Extraction and Sequencing

Total genomic DNA was extracted from two or three legs from each adult specimen or from muscle tissue from the abdomen (segments II–VI) of each larval (4th instar) specimen. The DNA was extracted using a DNeasy Blood & Tissue Kit (Qiagen, Hilden, Germany), following manufacturer instructions, and was then stored at −20 °C. A 658-bp fragment of *COI* was amplified using the AccuPower PCR PreMix kit (Bioneer Co., Daejeon, South Korea) and a previously published universal primer pair (LCO1490: 5′-GGTCAACAAATCATAAAGATATTGG-3′; HCO2198: 5′-TAAACTTCAGGGTGACCAAAAAATCA-3′) [34]. The 20 μL reactions were subject to the following conditions: 95 °C for 2 min; followed by 35 cycles of 95 °C for 30 s, 48–50 °C for 30 s, and 72 °C for 30 s; and a final extension step of 72 °C for 2 min. 

Amplification was confirmed by electrophoresis using 1.5% agarose gels and visualization under UV light, and the validated PCR products were then purified and sequenced using an ABI Prism Sequencer 3130 (Applied Biosystems, Foster City, CA, USA). The resulting sequences of both directions (reverse and forward) were aligned and edited using MEGA v.7 [35] and were checked manually to ensure correct alignment along codons. 

### 2.3. Genetic Diversity and Demographic History

The number of *COI* haplotypes (available on GenBank, ID: MZ573312-MZ573376) and genetic diversity indices, including haplotype diversity (*Hd*) and nucleotide diversity (π), were estimated using DnaSP v. 6.12.03 [36]. Meanwhile, the demographic signatures, selective histories, and both Tajima’s *D* [37] and Fu’s *Fs* [38] were calculated for each population using ARLEQUIN v. 3.5.2.2 [39], with population expansion signatures inferred using significance values that were calculated by generating 1000 random samples. Finally, to investigate the demographic history, pairwise mismatch distributions were inferred using a population growth-decline model in DnaSP v. 6.12.03 [36].

### 2.4. Haplotype Network Analysis

A minimum-spanning tree was generated using ARLEQUIN and then used to construct a haplotype network (HAPSTAR v.0.7), which was optimized using InkScape v. 0.92.4 (https://inkscape.org/ accessed on 19 July 2020). To gain a broader understanding of genetic relationships between populations from Vietnam and those from other parts of the world, the sequence dataset was also compared to other available *Ae. albopictus COI* sequences, which were identified using the basic local alignment search tool (BLAST) from GenBank (http://blast.ncbi.nlm.nih.gov/ accessed on 23 August 2020; Appendix A). All the compared sequences were trimmed to be totally matched with sequences in our study (658-bp fragment of *COI*).

### 2.5. Population Differentiation and Structure

Pairwise genetic differentiation (*F_ST_*) among populations was estimated using differences in haplotype frequencies, and sequential Bonferroni correction was used to adjust the significance levels in ARLEQUIN. 

To infer the geographic structure of *Ae. albopictus* populations in Vietnam, principal coordinates analysis (PCoA) of *F_ST_* values was performed, and GENALEX v.6.5 [40] was used for visualization. In addition, hierarchical analysis of molecular variance (AMOVA) was performed, using ARLEQUIN, to test several specific biogeographic hypotheses that relied on the spatial distribution of haplotypes, *F_ST_*, climatic separation [24] and urbanization classification [41]. Statistical significance was assessed according to the degree of differentiation among regions (Φ_CT_), among populations within regions (Φ_SC_), and within populations (Φ_ST_) using permutation tests with 1000 random permutations. Moreover, spatial analysis of molecular variance (SAMOVA) was applied to further determine the genetic and geographic clusters of homogeneous populations [42]. The number of clusters (K) was assessed by simulating a range of K values, from 1 (no genetic differentiation among all sites) to 16 (all sites are genetically differentiated from one another), with 1000 simulated annealing steps from each of the 100 sets of initial starting conditions. SAMOVA also generates *F*-statistics (Φ_SC_, Φ_ST_, Φ_CT_) using the AMOVA approach to maximize between-group variation.

### 2.6. Effect of Geography and Climate on Genetic Variation

Distance-based redundancy analysis (dbRDA) [43] was used to examine the effects of geographic distance and climatic factors on patterns of population genetic variation. The 16 sample sites were used as spatial units, and pairwise *F_ST_* values were used as a measure of genetic differentiation. A genetic distance matrix was used as a response variable, and geographic distance and climatic variables were used as explanatory variables. Geographic distance was transformed using third-order polynomials, and spatial variables that retained three spatial components were used for subsequent analyses. A total of 19 climatic variables related to temperature and precipitation, which are the most influential to this species’ biology and distribution [19], were obtained from the WorldClim database (www.worldclimate.com accessed on 23 August 2020). The most explanatory factors were selected, and collinearity variables were minimized using the principal component analysis (PCA) and Pearson’s correlation. Global dbRDA was then performed between *F_ST_* and selected spatial and climatic variables (Appendix A, Appendix A; Equation (1)).
*F_ST_* ~ climate (bio2+bio3+bio8+bio10+bio11+bio14+bio18+bio19)~ geography (geo1+geo2+geo3)          

The best model was selected using a stepwise approach that optimized the model by considering the Akaike information criterion (AIC). Condition tests were conducted to detect the individual effect of either climates or geographic distance on genetic variation. Variance components of dbRDA were then partitioned by running two models: (1) a partial model in which geography explains genetic data conditioned on climatic variables and (2) a reciprocal model in which climatic variables explain genetic data conditioned on geography [44]. The total variance (Inertia), proportion of the variation explained (% variation), and *P*-values are presented for each model.

The isolation-by-distance hypothesis was also tested using Mantel’s test based on the regression of pairwise *F_ST_*/(1 − *F_ST_*) on the natural logarithm (Ln) of straight-line geographic distance [45] (Appendix A). Mantel’s test and dbRDA analyses were performed using the VEGAN package in R [44].

## 3. Results

### 3.1. Genetic Diversity

*COI* sequences were generated for a total of 236 *Ae. albopictus* specimens in 16 localities in Vietnam (Table 1 and Appendix A). Fifty-nine of the 658 sites were variable (26 parsimonious informative and 33 singleton), and 65 haplotypes were identified, with a range of seven to 11 haplotypes per population (Table 1). The haplotype diversity was relatively high, with a mean of 0.9020 and range of 0.7714 (S06) to 0.9429 (S01), but estimates of nucleotide diversity were relatively low, with a mean of 0.0033 and range of 0.0019 (S06) to 0.0042 (S08; Table 1).

The evaluation of population expansion yielded significant negative values when all populations were pooled in a single data set (Tajima’s *D* = −2.3206, *P* < 0.001; Fu’s *Fs* = −26.8672, *P* < 0.001). All Fu’s *Fs* values, except that for S13, were also significant and negative. However, the Tajima’s *D* values were non-significant and negative values for all but four sites (S04, S06, S08, and S12; Table 1). The distributions of pairwise differences for the overall population were unimodal and the best fit of equilibrium distributions (Appendix A). Thus, the null hypothesis of a constant-size population under the neutral model was rejected for almost all localities. 

### 3.2. Haplotype Network

In the minimum-spanning network, the 65 haplotypes were separated by one to four mutational steps. Each of the three dominant haplotypes (H1, H3, and H5) was surrounded by closely related haplotypes, thus forming a ‘star-shape’ (Figure 1B), and a single-step mutation separated H1 from the two other dominant haplotypes (H3 and H5). The central haplotype (H1) accounted for 59 (25%) of the sampled specimens and was recovered from all the sample sites, except S04 and S05. The haplotype was also rare (one occurrence) in the specimens from S06, thereby making the haplotype H1 rare in the NE region. Meanwhile, H3 accounted for 31 (13.1%) of the sampled specimens and was totally absent from samples from the NE region (S04, S05, S06), and H5 accounted for 29 (12.3%) of the sampled specimens and was limited to northern Vietnam, with almost all individuals (*n* = 22, 75.9%) being sampled from the NE region. In addition, 45 (69.2%) of the 65 haplotypes were singletons, being represented by single specimens (Figure 1B).

Comparison to the 578 matching *COI* sequences obtained from GenBank revealed that the majority of the 65 haplotypes from Vietnam were novel, with only 15 being shared with sequences originating in other countries (Appendix A). However, it is worth noting that a large number of individuals (149/236) belonged to one of the shared haplotypes. Interestingly, the shared haplotypes included sequences that were associated with either temperature or tropical regions of the world. More specifically, H5 and its descendants (H15, H17, H18, H27, H30, and H36) were mainly derived from temperate localities (i.e., China, USA, Italy, Portugal, Russia, and Canada), whereas H1, H3, and their descendants (H13, H16, H40, H43, H50, and H59) were more often reported from tropical localities (i.e., Malaysia, Thailand, India, Singapore, Brazil, and Colombia; Appendix A). Therefore, it was concluded that the haplotype network included two genetically distinct groups, with H5 and its descendants belonging to a temperate group and H1, H3, and their descendants belonging to a tropical group (Figure 1B).

### 3.3. Genetic Differentiation and Population Structure

After correcting for multiple tests with the Bonferroni correction method, 66 out of 120 tests were statistically significant, with a mean *F_ST_* value of 0.0991 and range of −0.0333 to 0.4346, thereby suggesting some degree of structuring among the populations (Appendix A). The maximum difference was observed between S05 and S15, whereas the minimum difference was observed between S04 and S06. Interestingly, all comparisons among the NE populations (S04, S05, S06) were significant, with moderate to very high *F_ST_* values (0.1361 to 0.4346), whereas comparisons among the remaining populations yielded low and rarely significant *F_ST_* values (−0.0190 to 0.1647, Appendix A).

The spatial structure of *Ae. albopictus* populations in Vietnam is supported by several analyses. The distribution of haplotypes exhibited a latitudinal pattern and could be separated into three geographic regions (Figure 2A), which were characterized as containing mainly temperate haplotypes (Region 1), an admixture region with mainly tropical haplotypes (Region 2), and all tropical haplotypes (Region 3; Figure 2A). PCoA analysis consistently indicated a distinct structure between the NE (S04, S05, S06) and remaining populations (Figure 2B), and spatial structure was further supported by the SAMOVA analysis (Figure 2C). Indeed, Φ_CT_ values decreased when populations were divided into smaller groups (from *K* = 2 to *K* = 16), and the greatest Φ_CT_ value was detected when populations were clustered into two groups (*K* = 2, *P* < 0.0001). 

Furthermore, hierarchical analyses of molecular variance (AMOVA) also revealed significant geographic structuring among the 16 populations, with statistical support for almost all biogeographic hypotheses (Table 2). One strongly supported hypothesis was that the NE and remaining populations are significantly different (Φ_CT_ = 0.2914, *P* < 0.001). The separation of populations into three groups that corresponded to the three haplotype regions (Figure 2B) was also supported (Φ_CT_ = 0.1836, *P* <0.0001), as was the separation of populations into seven groups that corresponded to climatic regions (Φ_CT_ = 0.1599, *P* < 0.001). Finally, the separation of populations into three groups that were randomly selected among regions based on the urbanization level of each sample site (Φ_CT_ = 0.0323, *P* = 0.1329; Table 2) was not significant.

### 3.4. Effect of Geography and Climate on Genetic Variation

In comparison to the full model (Equation (1)), the best and final models were those that retained bio8 (mean temperature of wettest quarter), bio11 (mean temperature of coldest quarter), and geo3 (geographic component). 

The db-RDA of the best model indicated that geographic distance and climatic factors together explained 75.98% of the total genetic variation among *Ae. albopictus* populations (AIC: −30.93, *P* = 0.001; Table 3). Partial tests indicated that single climatic factors could explain more variance than geographic distance alone (64.55% compared to 33.73%), and when controlling the effects of geographic distance and climate, conditional testing further supported the conclusion that climate was a stronger predictor of genetic differentiation than geographic distance (42.25% compared to 11.42%; Table 3).

The ordination plot of the best model indicated that the mean temperature of the coldest quarter (bio11) was the most explanatory variable for the separation of NE populations (S04, S05, S06) from the other populations (Figure 3). In addition, the correlation between genetic and geographic distance (Mantel’s test) revealed a low but significant pattern of isolation-by-distance (Pearson’s coefficient of correlation [r] = 0.242, P 0.034; Appendix A).

## 4. Discussion

Even though *Ae. albopictus* is native to Vietnam and its impacts have been reported for several decades, the current study provides the first comprehensive investigation of the genetic variation and population structure of *Ae. albopictus* in Vietnam. The results reveal high genetic diversity with an admixture of temperate and tropical haplotypes, as well as an apparent population structure that is mainly determined by climatic differences. The current study’s findings are able to inform future vector control strategies in the country.

The genetic diversity of *Ae. albopictus* in Vietnam is relatively high (65 haplotypes) and is higher than almost all previous estimates for either native or invasive *Ae. albopictus* populations around the world (Appendix A), which likely reflects that Vietnam is one of the ancestral distribution areas of *Ae. albopictus*. For many species, the place of origin can be associated with the region of highest genetic diversity [46,47]. Moreover, the higher genetic diversity of populations in Vietnam could also be explained by our large population size collected across Vietnam because it is expected that species with larger geographic ranges will harbor more genetic diversity [48]. Indeed, we have sampled in many climatic regions in different landscapes (mountains, delta, and coastline) and in a large number of habitats in each locality (artificial and natural containers; Appendix A). Furthermore, the species’ genetic diversity could be augmented by the presence of multiple haplotype groups (i.e., temperate and tropical). Notably, the two haplotype groups observed in Vietnam (Figure 1B) correspond to the two most prevalent haplogroups in the world, specifically, the A1a (temperate region) and A1b (tropical region) haplotypes reported by Battaglia et al. [49] by the detection of diagnostic mutations at np 1820. The co-occurrence of two worldwide genetic clusters in Vietnam is in agreement with the result that has been reported in a population genomic study of *Ae. albopictus* [50]. Though both haplotype groups coexist, whether *Ae. albopictus* in Vietnam responds for source of both temperate and tropical regions globally is uncertain. However, considering the high affiliation between Vietnamese haplotypes to not only tropical haplotypes but also temperate haplotypes from many geographical regions worldwide (Appendix A) suggests the genetic data in this study could be useful for tracking the origin of the worldwide invasions. For instance, the previous studies have found a monophyletic group of the Brazilian population suggesting a single source from a tropical region in Southeast Asia, but the exact place was not tracked due to the insufficient sampling in the region [49,50]. Based on genetic data in the current study, we found two major haplotypes (H1 and H5) that have also been reported in Brazil (KX383924 and MK575475, respectively), whose similarities may suggest one of invasion pathways for *Ae. albopictus* populations in the country (Appendix A).

It is noteworthy that the occurrence of haplogroups marginally influenced the estimated genetic diversity in each region in Vietnam. Specifically, genetic diversity was greatest in the contact region (Region 2; *Hd* = 0.903, π = 0.0034), which harbored both the temperate and tropical haplotypes, followed by Region 3 (*Hd* = 0.8308, π = 0.0025), which harbored only tropical haplotypes, and then Region 1 (*Hd* = 0.760, π = 0.0022), which harbored mainly temperate haplotypes. The results of the present study suggest a new perception of the accessing level of genetic diversity in a certain region, which is not higher in tropical or temperate regions, but in a region of contact between them, such as the several northern localities (Region 2) of Vietnam (Figure 2, Table 1). It could be important information to identify the potential distribution of genetic groups of this species worldwide, especially in invaded regions, but larger samples from either native or invaded regions would be needed. However, all samples in this study were collected only in urban areas, and investigation in other landscapes including rural and mountainous areas may give a better evaluation of the genetic diversity level for the country. Furthermore, though the *COI* gene is widely used as an efficient genetic marker in many population genetic studies of *Ae. albopictus* to provide valuable insight into genetic variability in a novel area [9,10,11,21,51,52,53,54,55,56], further investigations utilizing other nuclear DNA markers such as simple sequence repeat (SSR) markers and single nucleotide polymorphism (SNPs) are encouraged to validate the present study’s results.

Interestingly, some previous studies revealed little or no differentiation between *Ae. albopictus* populations in native ranges such as in South-East Asia (China, Thailand, and Japan) using microsatellite loci [57] or studies using *COI* gene in northern Asia [54] or in Cambodia [10], the present study on the other hand provides multiple lines of evidence that support a geographic separation in *Ae. albopictus* populations in Vietnam. Particularly, the analyses in the current study clearly revealed a regional clustering of temperate and tropical haplotype groups that can be evidenced by the geographical distribution of haplotype groups (Figure 2A), and the PCoA plot (Figure 2B), SAMOVA tests (Figure 2C), and hierarchy AMOVA (Table 2). The population structure may reflect the current restriction of gene flow between temperate and tropical populations in Vietnam, in which the temperate haplotype group was found only above 19 °N, whereas the tropical haplotype group occurred more more in the southern regions (Figure 2A). Of note, the distinct population structure between temperate and tropical populations in the current study were separated by a single mutation, indicating a low genetic divergence among subpopulations. A plausible explanation for this phenomenon might be due to *Ae. albopictus* populations in each region having undergone a recent expansion range of major haplotypes, followed by the radiation of unique haplotypes. It can be clearly seen in the haplotype network that haplotypes H1 and H3 were dominant in almost all populations in Regions 2 and 3, whereas haplotype H5 was dominant in populations in Region 1, but in both cases, the dominant haplotypes were surrounded by many unique haplotypes (Figure 1B). 

It is obvious that geographical distance must be an important barrier to dispersal preventing genetic exchange of individuals between populations. Thus, previous studies have reported the effects of geographic distance on the genetic structure of *Ae. albopictus* populations [9,10,55]. However, the present study reveals for the first time the joint effects of both geographic distance and climatic factors on the genetic diversity of *Ae. albopictus* populations. Based on the results of dbRDA analyses, the observed variation was driven more by climatic factors (42.25%) than by geographical distance (11.42%; Table 3). The mean temperature of the coldest quarter (bio11) directly explained the distinct separation of the NE populations (S04, S05, S06) from the rest of the populations (Figure 3). The NE region is a particularly mountainous area (Table 1) that is annually affected by the northeast monsoon winds from China and that is bordered by the Gulf of Tonkin and Hoang Lien Son mountain range to the east and west, respectively [58]. As a result, the region occasionally experiences snowfall and hoarfrost freezing during winter and is colder and drier in winter than other areas within the same latitudinal belt [24,58,59,60]. Given the periodic harsh weather conditions in the NE region, *Ae. albopictus* populations are likely to undergo photoperiodic diapause to survive winter conditions, and the cold winter climate may cause an annual bottleneck event that prevents successful inhabitants of the non-diapause population in the NE region. Previous studies have demonstrated the unsuccessful long-term colonization of tropical strains of mosquito species *Wyemyia smithii* (Coquillett, 1901) that were translocated to temperate regions or of temperate populations that were subjected to tropical conditions [61,62]. The effect of cold temperature on the *Ae. albopictus* population was also reported by Zhong et al. [11], who found that the elimination of early introduced tropical genotypes by temperate ones within a period of 10 years of temporary colonization of tropical populations in Los Angeles, United States of America. Therefore, the genetic distinction between the NE region and the remaining region in this study is likely consistent with variation in ecophysiological traits (photoperiodic diapause and cold tolerance of eggs) since those traits are adaptive keys affecting both temporal and spatial abundance [63] and that enhance the adaptability of *Ae. albopictus* to colonize temperate environments [7]. Extensive studies of differentially expressed genes in various mosquito life stages has also revealed many distinctive differences in the gene expression and, thus, physiological processes of diapause and non-diapause populations [64,65]. It is noteworthy that some evidence has been revealed for the differences between northern and southern *Ae. albopictus* populations in Vietnam. For instance, *Ae. albopictus* specimens from northern Vietnam (Hanoi) are able to undergo diapause [66] and possess wing morphometrics that are more similar to those of specimens from other temperate areas (Korea, Japan) than to those of specimens from southern Vietnam [67]. The genetic structure related to ecophysiological traits (photoperiodic diapause) has also been reported between tropical and temperate regions in both invasive and native ranges worldwide [50]. However, future studies investigating the diapause ability of *Ae. albopictus* subpopulations in Vietnam are encouraged, perhaps using translocation experiments, to validate the findings of the present study.

Moreover, the maintenance of restricted gene flow by climatic factors could promote successful invasion from other environments by the effects of climate change (e.g., global warming) since adaptation under rapid environmental change is likely unnecessary [18]. In this context, non-diapausing tropical populations of *Ae. albopictus* are more likely to extend their ranges to the north because the effects of global warming could limit the range of the temperate zone [18]. In the present study, two haplotypes from the tropical group (H1 in S06 and H23 in S04) occurred in the NE region, which suggested an ongoing process of northward invasion, perhaps as a result of global warming. Surveillance on spatiotemporally genetic change of *Ae. albopictus* populations in wider ranges (both native and invasive regions) is encouraged to further assess the effects of climate change to the species’ genetic variation.

The findings of the present study regarding the population genetics of *Ae. albopictus* may improve predictions of vector-borne disease outbreaks and facilitate the development of dengue mosquito control in Vietnam. High levels of genetic diversity were observed in both the temperate and tropical *Ae. albopictus* populations and especially in the admixture population (Region 2; Figure 2A). Genetic diversity is considered a critical raw material for environmental adaptation and evolutionary capacity and promotes the establishment of new populations in uninhabited areas [18]. Therefore, the high genetic diversity of *Ae. albopictus* in Vietnam may allow them to resist changes in environmental conditions, pesticide resistance, or increase the likelihood of invasion success. A distinct population structure requires a specific control method for each subpopulation. Indeed, the success of biological control, in particular, requires the intensive selection of suitable biocontrol agents [7]. Moreover, the use of genetic modification to disrupt the diapause of mosquitoes could be effective for reducing temperate *Ae. albopictus* populations but would not affect populations in the tropics. The present study’s robust evidence for recent population expansion in Vietnam may indicate the consequences of urbanization [68], which has created diverse habitats for *Ae. albopictus* [69] or may suggest less efficient control methods [17]. 

## 5. Conclusions

Currently, the genetic variability among climatic regions of *Ae. albopictus* population in Vietnam is unknown. The present study of *Ae. albopictus* populations in Vietnam revealed high genetic diversity with evidence for the admixture of gene pools from temperate and tropical populations, thereby suggesting Vietnam could be an important original source for invaded areas globally. Interestingly, an uncovered population structure between these phenotypic strains of *Ae. albopictus* was found in a native range in Vietnam. The present study also revealed that the genetic variation observed among populations in Vietnam depended heavily on climatic factors, and to a lesser extent, geographic distance. These findings enrich the current understanding of the genetic diversity and population structure of *Ae. albopictus* in its native range and possibly allow for a better understanding of the worldwide invasion scenario. However, further studies conducted in larger areas and using different genetic markers are needed to validate our conclusions.

## Figures and Tables

**Figure 1 genes-12-01579-f001:**
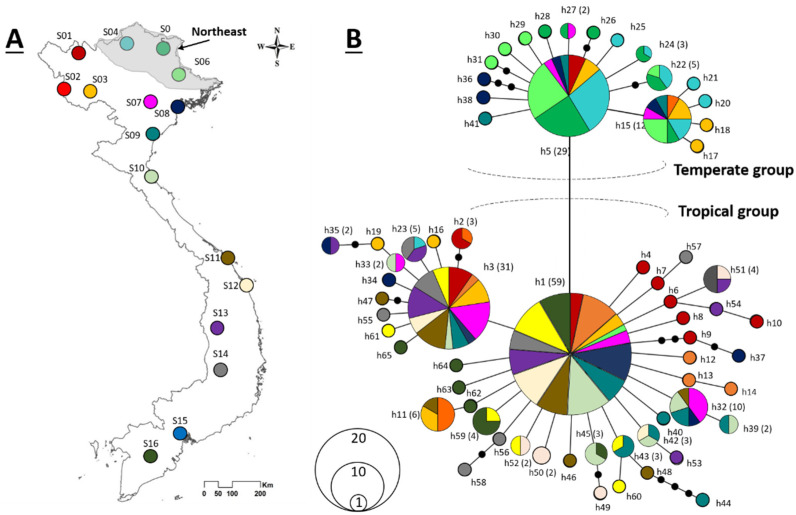
Map of collection localities (**A**) and phylogenetic network of *COI* haplotypes (**B**). (**A**) The color represents for each site matched with color in haplotype network. Codes for collection sites are shown in Table 1. (**B**) Haplotype network. Colors represent sampling localities (Table 1). Circles and colors represent haplotypes and correspond locality, respectively, and labels and numbers in parentheses indicate haplotype names and frequencies, when >1. Black dots represent intermediate haplotypes that were not recovered during the present study.

**Figure 2 genes-12-01579-f002:**
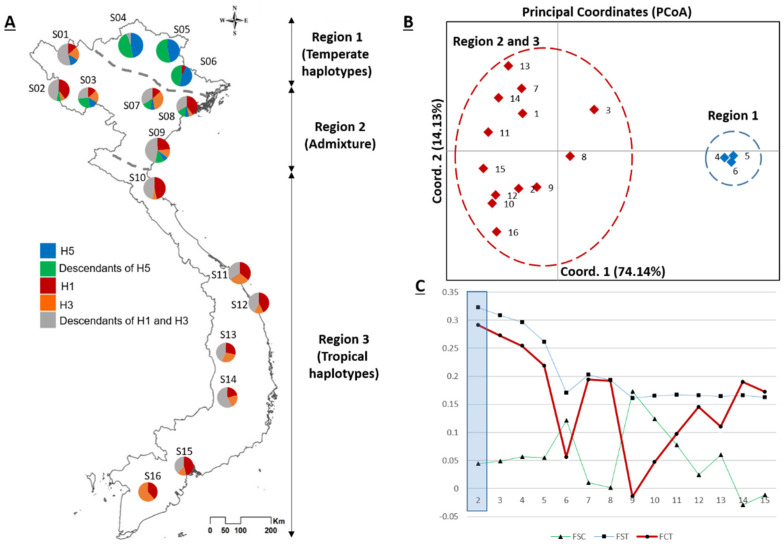
Distribution of *COI* haplotypes and population structure analyses of 16 *Aedes albopictus* populations in Vietnam. (**A**) Haplotype distribution map. Each pie chart illustrates the proportion of three major haplotypes and its descendants in each locality. The dashed lines represent the borders of the three regions indicated to the right of the map. (**B**) Principal coordinates analysis (PCoA) plot based on pairwise genetic distance. (**C**) Fixation indices obtained by SAMOVA for the best-clustering option at pre-defined values of K.

**Figure 3 genes-12-01579-f003:**
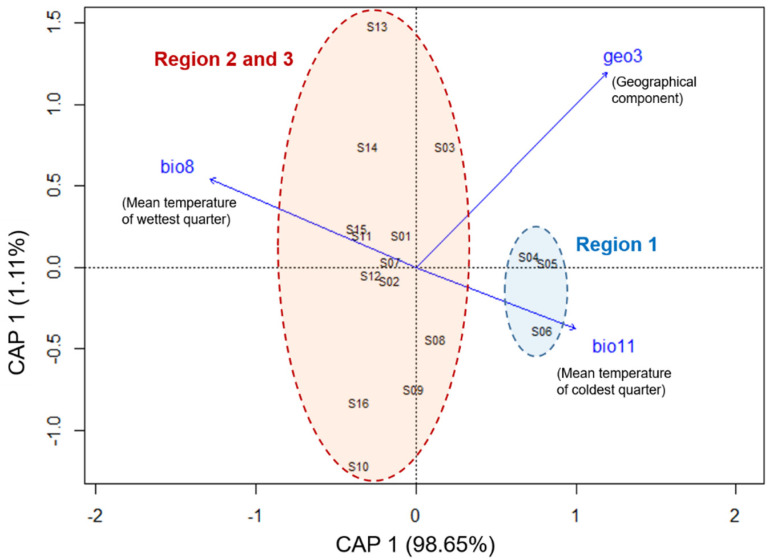
Plot of distance-based redundancy analyses from the final model. The dbRDA plot partitioned the genetic variance of *Ae. albopictus* populations explained by the predictor variables including two climatic variables, bio8—mean temperature of wettest quarter, bio11—mean temperature of coldest quarter and a spatial component, geo3—third-order polynomials transformation. Site codes are corresponded with definition in Table 1. Population clusters were indicated by colored circles, in which blue circle clustered populations from region 1 while red circle represent for group of populations from region 2 and 3. The first two axes explained 98.65% of the variability in the final model. Vector notations indicate the strength of the relationship between predictor variables and the dbRDA axes.

**Table 1 genes-12-01579-t001:** Location and genetic diversity of *Aedes albopictus* populations in Vietnam.

ID	CR	Locality	*N*	No. Haplotypes (N_H_)	Haplotype Diversity (*Hd*) (SD)	Nucleotide Diversity (π) (SD)	Tajima’s *D*	Fu’s *Fs*
S01	NW	Lai Chau province	15	10	0.9429 (0.0403)	0.0037 (0.0024)	−1.3221	**−5.1246**
S02		Dien Bien province	15	8	0.8286 (0.0849)	0.0030 (0.0020)	−1.3545	**−3.1153**
S03		Son La province	15	9	0.9333 (0.0397)	0.0036 (0.0023)	−0.1473	**−3.7633**
S04	NE	Ha Giang province	17	8	0.7794 (0.0985)	0.0025 (0.0021)	**−1.6227**	**−3.3956**
S05		Cao Bang province	15	7	0.7810 (0.1016)	0.0021 (0.0016)	−1.5728	**−3.0275**
S06		Lang Son province	15	7	0.7714 (0.1001)	0.0019 (0.0016)	**−1.7724**	**−3.3800**
S07	ND	Hanoi city	15	7	0.8381 (0.0680)	0.0024 (0.0016)	0.5623	**−2.6767**
S08		Hai Phong city	15	10	0.8571 (0.0901)	0.0042 (0.0026)	**−1.5855**	**−4.4915**
S09	NC	Thanh Hoa province	17	11	0.9338 (0.0426)	0.0032 (0.0021)	−1.5294	**−6.8099**
S10		Nghe An province	15	7	0.7810 (0.1016)	0.0020 (0.0014)	−1.1519	**−3.4900**
S11	SC	Da Nang city	14	7	0.8242 (0.0781)	0.0025 (0.0017)	−1.3494	**−2.7610**
S12		Quang Ngai province	14	7	0.8132 (0.1158)	0.0023 (0.0017)	**−1.7663**	**−2.9501**
S13	CH	Gia Lai province	14	7	0.8571 (0.0652)	0.0031 (0.0020)	−1.3491	−1.9899
S14		Dak Lak province	14	8	0.9121 (0.0440)	0.0032 (0.0021)	−0.6081	**−3.1394**
S15	S	Ho Chi Minh city	13	7	0.7949 (0.1091)	0.0020 (0.0015)	−1.2441	**−3.8060**
S16		Can Tho province	13	7	0.8333 (0.0861)	0.0020 (0.0015)	−1.5939	**−3.7332**
		Total	236	65	0.9020	0.0033	**−2.3260**	**−26.8672**

ID, locality code used for the downstream analyses; CR, climatic region; N, number of individuals analyzed; significant values of neutrality tests (Tajima’s *D* and Fu’s *Fs*) are indicated in bold.

**Table 2 genes-12-01579-t002:** Analysis of molecular variance (AMOVA) testing of hypotheses for the partitioning of genetic variation among *Ae. albopictus* populations in Vietnam.

Hypothesis	Source of Variation	Percent of Variation	F-Statistic	*P*-Value
Two groups: PCoA plot and haplotype distribution map - Northeast (S04, S05, S06) - The other populations.	Φ_CT_	29.14	0.2914	<0.001
	Φ_SC_	3.15	0.0445	<0.001
	Φ_ST_	67.71	0.3229	<0.001
Three groups: spatial distribution of haplotypes (Figure 2A) - Region 1 (S04, S05, S06) - Region 2 (S01, S02, S03, S07, S08, S09) - Region 3 (S10, S11, S12, S13, S14, S15, S16)	Φ_CT_	18.36	0.1836	<0.001
	Φ_SC_	2.71	0.0302	<0.01
	Φ_ST_	78.93	0.2107	<0.001
Seven groups, geography and climate - Northwest (S01, S02, S03) - Northeast (S04, S05, S06) - North Delta (S07, S08) - North Central (S09, S10) - South Central (S11, S12) - Central Highlands (S13, S14) - The South (S15, S16)	Φ_CT_	15.99	0.1599	<0.001
	Φ_SC_	1.45	0.0173	0.06158
	Φ_ST_	80.25	0.1744	<0.01
Three groups: level of urbanization - Class I, municipalities (S7, S8, S11, S15, S16) - Class II, provincial cities (S1, S2, S3, S4, S5, S6, S13, S14) - Class III, provincial towns (S9, S10, S12)	Φ_CT_	3.23	0.0323	0.1329
	Φ_SC_	13.88	0.1434	<0.001
	Φ_ST_	82.89	0.1711	<0.001

Φ_CT_: among groups; Φ_SC_: among populations within groups; Φ_ST_: within populations.

**Table 3 genes-12-01579-t003:** Effect of geographic and climatic factors on *COI* sequence variation.

Variable	Inertia	Percent of Variation (%)	*P*-Value
*Global dbRDA*			
Geographic distance + Climates	0.2961	75.98	0.002
*Partial dbRDA*			
Climates	0.2516	64.55	0.001
Geographic distance	0.1314	33.73	0.010
Climates | Geography	0.1646	42.25	0.002
Geography | Climates	0.0445	11.42	0.020

## Data Availability

Not applicable.

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
