# Peer review of "Genetic Diversity and Population Structure of the Asian Tiger Mosquito (Aedes albopictus) in Vietnam: Evidence for Genetic Differentiation by Climate Region"

_genes, 2021, doi:10.3390/genes12101579_

Round 1

Reviewer 1 Report

The article "Genetic diversity and population structure of the Asian tiger mosquito (Aedes albopictus) in Vietnam: evidence for genetic  differentiation by the climate regions” is an interesting study on genetic and spatial differentiation of populations of Aedes albopictus within its native range. Such data are highly needed to understand better the process of worldwide spread the species has undergone as well as its potential to adapt to different eco-climatic conditions. Indeed information on populations from the native range are often missing, hampering the possibility to infer the source populations in the invasive range. The study appears well performed and described, with a good sampling and sequencing effort across the whole country and a thorough analysis of COI haplotypes in comparison with what observed by previous studies.

Line 74: I would suggest to refer also to CHKV, since Ae. albopictus  appears to be a crucial vector for it.

Minor comments:

line 95:  it would be helpful to add somewhere information on ecoclimatic regions; at least in table S1 or possibly also in Figure 1; as an alternative it would be helpful to ad Figure 1.A an indication of at least the distinction in temperate and  subtropical/tropical group.

Line 146: how long where finally compared sequences?

Line 347-349: please reformulate

Line 372: correct in divergency

Line 382: please reformulate

Line 419-421: please reformulate

Line 422-424: I guess we could say the same about the possible extension of temperate climate regions towars north, don’t you think so?

Line 423-425: I think this is very speculative.

Line 430: what do you mean with “vector outbreaks”?

Line 445-447:  To what results in particular do you refer with this sentence? The article contains a lot of good results but in which way you think they give you information on the importance of human-aided transport in the dispersal of Ae. albopictus?

Line 452: insert “of” before “gene pools”

Line 453: while I’m sure based on previous publications that Vietnam is one important source for the worldwide spread of the tiger mosquito I do not know how the genetic diversity  you find in this article or the admixture between gene pools from temperate and tropical areas can prove this. This seems a bit oversimplified.

Author Response

Dear Reviewer 1,

Thanks indeed for your kind comments and suggestions. We have corrected and revised our manuscript based on your comments and suggestions.

The files of responses and revised manuscript are attached.

Thanks again for your assistance.

YJ Bae

Reviewer 2 Report

The present work by Van Duong et al. address a very important and yet often overlooked question of what is the extent of genetic variation that is present in any given species of mosquito and how does that similarity or variation effects its disease transmissible capacity. The authors have done an excellent job of doing a wonderful work of collecting all these mosquitoes and analyzing quite thoroughly. But there are some issues that need to be addressed. Most of these changes are with respect to the English language, the methodology is sound. I have listed my concerns below:

  1. The abstract needs a rewriting. For ex. he first two sentences both need to be restructured to be understood correctly by all readers. It is same for the sentence in line#20-22 and the last sentence.
  2. Line 17 at=from
  3. line 18 explain=explained
  4. line 21 a=in a
  5. line 14-15 could be stated as "The present study aims to investigate the genetic diversity and population structure of Aa. albopictus across various climatic regions of Vietnam".
  6. line 37 starting at "interestingly,..." please rewrite the sentence.
  7. line 41 delete "in a region"
  8. line 65: why you have used the term "likely" here? is there no weather data to talk about it affirmatively?
  9. line 67 delete intensively
  10. line 79 not conducted=not been conducted
  11. line 116 larva=larval
  12. line 178: PCA-first time used of abbreviation.
  13. line 195 & 196: clarify what "sites" mean in two different locations, they don't mean the same. One is geographical, and the other is genetic. It is confusing.
  14. The discussion part has many sentences that need serious restructuring/rewriting. I will state a couple of examples below, please look at other sections as well: line 320-322 starting at "Indeed..."; line 338-340 starting at "Based on...". There are many like this, and I can't list them all here. Please check on all those.
  15. line 334 suggesting=suggests
  16. line 358 SNP-first time use of abbreviation
  17. line 364-367 starting at "Particularly,..."-this looks like an incomplete sentence. Please rewrite it.
  18. line 372 divergent-divergence
  19. line 382 delete is
  20. line 382 clearly reveal-reveals (no need of clearly)
  21. line 383 geographic-geographical
  22. line 383 features-factors
  23. line 383 variation-diversity
  24. line 385 temperate-temperature
  25. line 402 America-USA or United States of America
  26. line 402 genetically-genetic
  27. line 418 crosses-something more appropriate like breeding or other alternate terms.

Like you can see above, most of the issues with this MS is not technical, it is very well done. So, please go through your MS thoroughly to make sure it sounds grammatically just fine when published.
